# Effect of Thidiazuron on Terpene Volatile Constituents and Terpenoid Biosynthesis Pathway Gene Expression of Shine Muscat (*Vitis labrusca × V. vinifera*) Grape Berries

**DOI:** 10.3390/molecules25112578

**Published:** 2020-06-02

**Authors:** Wu Wang, Muhammad Khalil-Ur-Rehman, Ling-Ling Wei, Niels J. Nieuwenhuizen, Huan Zheng, Jian-Min Tao

**Affiliations:** 1College of Horticulture, Nanjing Agricultural University, Nanjing 210095, China; 2017204015@njau.edu.cn (W.W.); muhammad@njau.edu.cn (M.K.-U.-R.); 2016104029@njau.edu.cn (L.-L.W.); 2The New Zealand Institute for Plant and Food Research Ltd. (PFR), Private Bag 92169, Auckland 1142, New Zealand; Niels.Nieuwenhuizen@plantandfood.co.nz

**Keywords:** grape, thidiazuron (TDZ), aroma compounds, Shine Muscat

## Abstract

Volatile compounds are considered to be essential for the flavor and aroma quality of grapes. Thidiazuron (TDZ) is a commonly used growth regulator in grape cultivation that stimulates larger berries and prevents fruit drop. This study was conducted to investigate the effect of TDZ on the production of aroma volatiles and to identify the key genes involved in the terpene biosynthesis pathways that are affected by this compound. Treatment with TDZ had a negative effect on the concentration of volatile compounds, especially on monoterpenes, which likely impacts the sensory characteristics of the fruit. The expression analysis of genes related to the monoterpenoid biosynthesis pathways confirmed that treatment with TDZ negatively regulated the key genes *DXS1*, *DXS3*, *DXR*, *HDR*, *VvPNGer* and *VvPNlinNer1*. Specifically, the expression levels of the aforementioned genes were down-regulated in almost all berry development stages in the TDZ-treated samples. The novel results from the present study can be used to aid in the development of food products which maintain the flavor quality and sensory characteristics of grape. Furthermore, these findings can provide the theoretical basis that can help to optimize the utilization of TDZ for the field production of grapes at a commercial scale.

## 1. Introduction

Grape (*Vitis vinifera* L.) is a high value fruit crop worldwide which is consumed fresh as berries or utilized to produce wine. Shine Muscat, a table grape cultivar which originated from Japan, is currently very popular in Asia—especially in China and Japan, due to its high brix level, large berry size and pleasant Muscat flavor [1]. The taste and aroma of grape are affected by many components such as soluble sugars, organic acids and volatile compounds. Among them, aroma is the key factor affecting the flavor of grapes. Aroma compounds can exist as free volatiles and a subset can also occur as glycoside conjugates [2], which collectively contribute to the sensory characteristics of grapes. In previous studies, the volatile composition of Shine Muscat grape has been extensively documented and found to be primarily derived from C6 compounds, alcohols, esters, aldehydes and terpenes [3,4].

A large number of volatile compounds have been identified in different grape cultivars and fractions of these compounds have been characterized as the main flavor component of fruit based on their quantitative abundance and olfactory thresholds [5]. In Muscat grapes, terpenes are considered to contribute to floral and fruiting characters and eventually control the Muscat flavor of berries [6]. Terpenoids are derived from C5 isoprene units, while the biosynthesis of monoterpenoids always proceeds through the methyl-erythritol-phosphate (MEP) pathway which occurs in the plastids. The biosynthesis of sesquiterpenes always occurs via both the mevalonic acid (MVA) pathway in the cytosol and the methyl-erythritol-phosphate (MEP) pathway in the plastids [7,8]. Terpenoid concentrations are changed in response to alterations in the expression of genes involved in the MEP and MVA pathways, which still remain to be identified and functionally characterized in grape [9].

The term “aroma compounds” refers to the detection/smelling of “volatiles” by the nose, whereas, the term “flavor” refers the coupling of both aroma and taste. Glycosides influence the aroma profiles of berries and can serve as the precursors for the free fraction of volatiles. Bound volatiles can be released during grape processing and fermentation through contact with enzymes such as glycosidases. During wine storage, they can also be chemically hydrolyzed under acidic conditions. In table grape, the aroma glycoside pool may influence the aroma profiles of berries as they can serve as the precursors for the free fraction of volatiles that may be released by saliva enzymes in the mouth or by the plant enzymes released during consumption [10,11,12]. The bound fraction of monoterpene aromas can also contribute to the production of the final monoterpene profiles of value-added products such as juice and wines [13,14].

The berries of Shine Muscat are relatively small in size and easily fall off the vine under natural conditions in the field, resulting in a negative impact on the harvestable yield and fruit quality [1]. Therefore, in order to cultivate berries with more commercially relevant fruit characteristics, treatments with gibberellic acid (GA_3_) and thidiazuron (TDZ) can be applied at one or more times at different stages of grape berries during their fruit development [15,16]. GA_3_ has been widely used in the cultivation of table grapes due to its multifunctional ability, including the prevention of fruit drop, berry enlargement and the development of seedless fruit [17]. TDZ is a novel phenylurea plant growth regulator with dual auxin and cytokinin like activities, which are capable of inducing cell division and promoting fruit enlargement, thus improving the fruit quality and commodity values [18,19]. In apple, treatments with TDZ have been shown to alter the apple shape index and cause a reduction in the red surface of fruit and TDZ treatment also increased the percentage of fruit that were asymmetrical and exhibited calyx-end rot [20]. Plant endogenous hormone levels are regulated by TDZ and floriculture applications have shown a benefit for the inhibition of leaf yellowing of cut stock flowers by protecting leaf chlorophyll from degradation after transfer to vases [21]. Similarly, the immersion of grape berries in a TDZ solution resulted in higher quality fruit and increased firmness of fruit flesh [22]. Collectively, these findings show promise for using TDZ in grape cultivation and warrant additional study for the utility of this compound at a commercial scale.

In recent years, researchers focused on the effect of the exogenous application of hormones, environmental conditions and cultural practices on the volatile compounds of grape berries [23,24]. Limited studies have been conducted to characterize the effect of TDZ application on the volatile compounds, physicochemical composition and taste evaluation of grape berries—especially volatile monoterpenoids [25,26,27,28]. The novel work presented in this study was performed to explore the effect of TDZ application on the production of aroma volatile compounds in grape under field conditions and to identify the key genes involved in the terpenoids biosynthesis pathways that were suppressed by the elicitors.

Findings from this study will help to provide a deeper understanding of the volatile composition after TDZ application in Shine Muscat berries, and they will ultimately serve as a foundation for future improvement of berry flavor and overall quality of table grapes and also to establish the relationship between volatile levels, berry flavor and consumer preference.

## 2. Results and Discussion

### 2.1. Physicochemical Parameters of Grape Berries Altered by TDZ

Grape berries were harvested from treated (TDZ treated) and control (CK) plants during the pre-veraison (12 July), veraison (28 July) and harvest stages (28 August) during the 2018 and 2019 growing season and subjected to determine the physicochemical parameters. As shown in Figure 1, in 2018, the application of TDZ significantly increased the berry mass in treated samples (all three stages) in comparison with CK. The TA content, in treated samples collected during the stage of 28 July 2018 and 28 August 2019, was significantly lower; meanwhile, the TSS concentration exhibited a decreasing trend in the treated berries during the veraison and harvest stages. The three stages showed comparable physicochemical data across the two constitutive years with no differences in phenology. From the picture of individual grapes (Figure 1 bottom), the color of the large and round TDZ-treated berries maintained a darker green appearance compared with the smaller and more oval CK, and the same effect was observed in samples collected in 2019 (data not shown). In accordance with previous reports [19,29], the data in this study pertaining to TA, TSS and other physiological parameters suggest that the TDZ treatment might play a major role in delaying the maturity of berries and a resultant reduction in the berry sweetness during the harvest stage.

### 2.2. Free Volatile Monoterpene Compounds from Berries Treated with TDZ or CK

Aroma compounds can exist in a free volatile form as well as a glycosidically bound/odorless form in grape berries. The Muscat berry aroma contains higher levels of both free and bound aroma compounds [30]. Monoterpenes are the main volatile compounds in Shine Muscat grape berries, which are responsible for producing the characteristic Muscat flavor of berries [3,31]. Results presented in this work determined the concentrations of the free and bound monoterpenes of the control and TDZ-treated berries collected at the time of harvest during both sampling periods (2018 and 2019) (Table 1). For the free terpene volatile fraction, a total of 13 compounds were identified and quantified. In 2018 and 2019, the total volatile compounds in CK were generally 3- and 4-fold greater than the TDZ-treated berries, respectively. For the two consecutive years of study, the most abundant free monoterpene aromas in berries were linalool, geraniol and d-limonene. During both years of study, almost all of the monoterpenes showed a significant decrease in berries that were treated with TDZ. Among these, linalool, geraniol and d-limonene were the most odoriferous and abundant monoterpenes. Specifically, the respective reduction of these compounds in the 2-year study of the TDZ-treated berries were as follows: 68%, 89% and 69% (2018), and 74%, 96% and 73% (2019). After treatment with TDZ, a remarkable decrease was also recorded for α-Terpineol, Citronellol, β-Myrcene, α-Citral and Neral. These compounds were also detected in other grape cultivars and are known to influence the Muscat flavor of berries [4]. Among the monoterpenes (Table 1), only the concentration of citronellal increased after the TDZ treatment, which is most likely due to the conversion of citronellal to citronellol. Importantly, these findings were in agreement with previous reports, where the most abundant monoterpenes were characterized in Shine Muscat berries [4,31] and monoterpenes concentrations of berries were reduced after application with a TDZ-like hormone [3].

Many volatile aroma compounds have been identified in grape berries but the predominate compounds contributing to the aroma profiles of Muscat grape berries were represented by the monoterpenes [5]. Interestingly, almost all of the major free monoterpenes’ volatiles detected in our studies were found to be lower in the TDZ-treated berries compared with those treated with CK. The lower levels of monoterpenes in the TDZ-treated berries may eventually result in a lower quality in the flavor and aroma of the grape. To the best of our knowledge, limited studies have been conducted that measure the aroma of TDZ-treated table grapes during fruit development. Previously, the mode of action of TDZ was compared to CPPU (N-(2-Chloro-4-pyridyl)-N′-phenylurea), which shares a similar functional role for reducing the complexity and contents of fruit volatiles [32,33].

### 2.3. Bound Fraction of Monoterpenes Compounds from Berries Treated with TDZ or CK

In the present study, a total of 16 glycosylated aroma compounds were identified, among which, myrtenol, (*E*)-furanoid linalool and (*Z*)-furanoid linalool were differently identified compounds to compare with free aroma compounds. In comparison with the free monoterpene volatile compounds, which primarily consisted of linalool, geraniol and d-limonene, the bound monoterpene profiles were markedly different. The highest concentrations found were geranic acid, (*E*)-furanoid linalool and (*Z*)-furanoid linalool, all of which were detected in higher amounts in the bound fraction compared with the free fraction (Table 2). In accordance with previous studies [34], the total concentrations of bound monoterpenes were generally higher than the free fractions of monoterpenes.

It is important to note that almost all of the concentrations for bound monoterpenes recorded in CK were higher than those observed from the TDZ treatment. Similar findings were also observed in the free fraction of monoterpene volatiles. In the bound fractions, the monoterpene linalool was higher than what was detected from the free fraction, while geraniol and d-limonene in free form were also detected at lower levels compared with the bound fractions. Specifically, the highest concentration among the three free volatile compounds was geraniol (218.12 ng/g FW, in 2019, CK), followed by linalool (137.74 ng/g FW, in 2019, CK) and d-limonene (122.41 ng/g FW, in 2019, CK). During the course of both study years, the concentration of these compounds showed approximately a 4-fold decrease subsequent to treatment with TDZ both for free and bound terpenoids.

### 2.4. Effect of TDZ on Grape Terpene Synthase (TPS) Activity

To characterize the functional role of TDZ in relation to the activity of plant terpene synthases (TPSs), we compared the total TPS enzyme activity of treated and control berries during various stages of development on 12 July, 28 July and 28 August in 2018 and 2019 by using the plant terpene synthase (TPS) ELISA Kit. TPS concentrations were determined using the formula y = 0.0026x + 0.1466, where x represents the TPS activity, and y indicates the absorbance of the sample solution (Appendix A). As shown in Figure 2, our results revealed that in the pre-veraison (12 July) and veraison (28 July) stages, the application of TDZ resulted in a significant decrease in TPS activity. During the harvest stage, TPS activity in treated and control berries was nearly equivalent. The activity of TPS was found to increase in accordance with the progression of berry development.

Terpene synthases are the type of enzymes that regulate terpenes biosynthesis, using the initial substrates (GPP, FPP and GGPP) build of the simple C5-unit isopentenyl pyrophosphate and its isomer dimethylallyl pyrophosphate (DMAPP), which play an important role in terpene metabolism [35]. Over the course of the past three decades, there has been extensive characterization and classification of TPS enzymes which have divided them into seven subfamilies: TPS-a, TPS-b, TPS-c, TPS-d TPS-e/f, TPS-g and TPS-h [36]. Thus, it is highly likely that different types of TPS enzymes catalyzed the observed mixtures of monoterpene volatiles that were released by the grape berries in our study. The majority of the functionally characterized TPSs are multiproduct enzymes and the specific functions of the TPS families have been described in multiple papers [36,37]. Under our experimental conditions, TPS activity during the harvest stage of berry development did not exhibit an obvious difference between the TDZ treatment and control (Figure 2). These observations help to support the hypothesis that the TDZ activity may affect the development of berry aroma development in the early stages—especially during the pre-veraison and veraison stages.

### 2.5. Effect of TDZ on Gene Expression of Berries

Monoterpenes were the most abundant class of compounds detected in Shine Muscat berries. In order to characterize the monoterpenoid metabolic pathway at the gene expression level, we performed targeted transcript analyses by RT-qPCR. The monoterpenoid metabolism pathway has been previously characterized in different grape species. Critical steps for monoterpene biosynthesis have been revealed within the MEP pathway [38], and as a result, we focused our studies on the genes within this pathway, in addition to the relevant and functionally characterized TPS genes that act downstream of the MEP pathway [39]. Specifically, we selected the 1-deoxyxyulose-5-phosphate synthase (*VvDXS1* and *VvDXS3*), 1-deoxy-d-xylylose-5-phosphate reductoisomerase (*VvDXR*) and (*E*)-4-Hydroxy-3-methylbut-2-enyl diphosphate reductase (*VvHDR*) genes due to their considerations as the key enzymes responsible for controlling the flux of the MEP pathway [40]. In addition, we also selected the *VvPNGer* and *VvPNlinNer1* genes, which were confirmed as the highest expressed *TPS* genes leading to the biosynthesis of monoterpenes such as linalool and geraniol. During the two consecutive years of investigation, the expression profiles of most of these selected genes showed a decrease after the treatment with TDZ compared with the control.

From Figure 3 and Figure 4 (2018 and 2019, respectively), the *VvDXS1* transcript abundance was significantly lower in the TDZ-treated berries on 28 August (harvest stage), which represented the time point where berries were at a commercially relevant maturity level. On the other hand, no significant differences in the *VvDXS1* expression were observed prior to the mature berries. The *VvDXS3* expression was significantly lower in the TDZ-treated samples during two developmental stages (i.e., veraison and maturity stages). These results are in good accordance with two separate reports which identified an association between *VvDXS1* and *VvDXS3* and the content of monoterpenoids in *Vitis* spp. [41,42]. The *VvDXR* and *VvHDR* genes showed an obvious reduction in transcript abundance after the first application of TDZ and also throughout the last three time points of the sample collection—including the pre-veraison, veraison and commercially mature stages. Additionally, *VvHDR* overall was the highest expressed MEP pathway gene. In “Gewürztraminer” berries, the accumulation of the *VvHDR* transcript exhibited a good correlation with the accumulation of terpenoids [43]. However, in Riesling berries, light treatment did not have any effect on the *VvHDR* expression, and the accumulation of the *VvHDR* transcript was not significantly correlated with the accumulation of linalool or other monoterpenes [44]. In our study, a 5-fold increase in expression was observed for the *VvDXS3* gene during fruit development, making it the most responsive gene affected by the TDZ treatment observed under our experimental conditions. These data are in agreement with the hypothesis that *DXS* is the first-rate limiting enzyme for the biosynthesis of plastidial monoterpenes [45].

The initial substrates for the biosynthesis of monoterpenoids are C5-unit isopentenyl pyrophosphate and its isomer dimethylallyl pyrophosphate. These substrates are converted to GPP by geranyl diphosphate synthases (GPPS). Terpene synthases are subsequently responsible for catalyzing the formation of monoterpenes from GPP. To provide an additional characterization of this biosynthetic pathway, two *TPS* genes involved in linalool and geraniol formation were specifically selected for transcript analyses [37,46,47], revealing nearly identical patterns of gene expression. At the first time point of the treatment with TDZ, the expression levels of *VvPNGer* and *VvPNlinNer1* were slightly higher in treated berries in comparison with CK in both years. After the second time point of treatment, the expression of *VvPNGer* and *VvPNlinNer1* was significantly higher in the control samples compared with the TDZ-treated samples, at both the pre-veraison and veraison stages. At the commercial maturity stage, the expression of *VvPNlinNer1* was higher in the TDZ-treated berries as compared with controls; thus, from two weeks post-application until pre-harvest, treatment with TDZ results in a decrease in the expression of *TPS* genes. Among the *TPS* genes, *VvPNLinNer1* was identified as the most highly expressed gene in all samples and its expression was the most reduced by TDZ at the veraison stage. These data were in agreement with previously published results showing that the expression of *VvPNLinNer1* and other TPS genes were most influenced by hormone treatment during the beginning of the berry color changing stages [48].

In our study, treatment with TDZ significantly decreased the concentrations of volatile monoterpenoids. Collectively, these compounds contribute to the Muscat flavor of berries, and the MEP related genes and downstream TPS genes and enzymes that contribute to the biosynthesis of monoterpenoids are all negatively affected by treatment with TDZ. We conclude that TDZ functions as a potent chemical that affects volatile compounds by reducing the gene expression patterns of several monoterpene biosynthesis pathway genes as well as the TPS enzyme activity in Shine Muscat berries.

## 3. Materials and Methods

### 3.1. Plant Material and Field Conditions

The present study was conducted during 2018 and 2019 in an experimental vineyard of Nanjing Agricultural University, Nanjing, China. Six-year-old Shine Muscat (*Vitis labruscana* Bailey × *V. vinifera* L.) grapevines were used in the study. The materials were planted in an “H” shape and grown under sheltered tunnels in sandy soil, with a spacing of 8 m within rows and 6 m between rows. Clusters of fruitlets were randomly selected and divided into two groups. The first group of clusters were treated with 25 mg·L^−1^ GA_3_ and 5 mg·L^−1^ TDZ (treated), while the second group of clusters were treated with 25 mg·L^−1^ GA_3_ as a control (CK). Tween 80 was used as the wetting agent (0.1% *v/v*). All fruitlets were treated 2 times, with the first treatment commencing at flowering and the second treatment occurring two weeks after flowering. The flowering date and subsequent fruit development rate were carefully monitored during both years and there were no discernable differences in flowering time nor the fruit development rate between the two years, therefore, equivalent sampling dates were the same across both years. This was reflected in the very similar phyciso-chemical data across both years (Figure 1) with no statistical differences apparent.

### 3.2. Sampling and Physicochemical Analysis

Berries for gene expression were collected at 24 h after treatment and the specific stages of sampled berries were pre-veraison, veraison and commercial maturity. The corresponding sampling dates were 12 May, 26 May, 12 July, 28 July and 28 August, in 2018 and 2019. Samples taken from the different stages were accordingly as follows: 12 May 2018, 26 May 2018, 12 July 2018, 28 July 2018 and 28 August 2018, and 12 May 2019, 26 May 2019, 12 July 2019, 28 July 2019 and 28 August 2019, respectively. The samples which were harvested on 28 August were also used for the determination of the average berry mass (ABM), total soluble solids (TSS), titratable acidity (TA) and the free or bound fraction of aroma compounds. Berries with different treatments were collected at different stages and instantly frozen in liquid nitrogen and stored at −80 °C until further use. Total soluble solids (TSS) were determined using an Abbé refractometer (type Rx-5000; Atago, Tokyo, Japan). Titratable acidity (TA), which was expressed as grams of tartaric acid per 100 g of fresh weight, was determined by titration with 0.1 N NaOH to a final pH of 8.1 using an automatic titration system [27].

### 3.3. Extraction of Free Fraction Aroma Compounds

The extraction of the aroma compounds was carried out as previously described by [49] and three biological replicates of samples were prepared. In each replicate, 40 g of sample were taken and placed into vials containing 0.5 g d-Glucosidase and 1 g PVPP (Polyvinylpyrrolidone). The samples were then ground with an immersion blender (Ultra-Turrax, IKA, Staufen, Germany). Afterwards, 30 g of fine mixture was transferred into a 50 mL centrifuge tube and placed at 4 °C for 2 h. After centrifugation at 5000× *g* for 10 min, 8 mL of the supernatant was collected and transferred to a 20 mL SPME vial (tightly capped with a PTFE–silicon septum, with a magnetic stirrer) containing 1.5 g NaCl.

### 3.4. Isolation of Bound Fraction Aroma Compounds

The isolation of the aroma precursors was performed by adsorption on Cleanert PEP-SPE resin (150 mg/6 mL; Bonna-Agela Technologies, Tianjin, China), as previously described [30,31]. Prior to this step, the resin was equilibrated with pure water and 95% methanol (10 mL each). Five ml of cleared berry juice was passed through the Cleanert PEP-SPE column and the water-soluble compounds were eluted with 5 mL of water. The free volatiles were eluted with 20 mL of dichloromethane and the aroma precursors with 20 mL of methanol at a flow rate of approximately 2.5 mL/min. The methanol eluent was concentrated to dryness by a rotary evaporator under vacuum and then redissolved in 5 mL of a 2 M citrate–phosphate buffer solution (pH 5.6). Subsequently, 500 µL of Rapidase AR 2000 (Aspergillus niger, USA) enzyme solution (100 mg/mL in 2 M citrate–phosphate buffer, pH 4.6) was added to the glycoside extract and the mixture was oscillatory vortexed. Optimum conditions of enzymatic hydrolysis were performed according to [50] in a sealed tube at 40 °C for 16 h to liberate the free aromas. The released compounds were extracted by solid-phase microextraction (SPME) [51] (Section 3.5).

### 3.5. SPME-GC-MS Analysis of Aroma Compounds

The determination of the free and liberated bound aroma compounds was carried out following the methodology described by Wu et al. [4], with a slight modification. A 50/30 μm polydimethylsiloxane/divinylbenzene/carboxen (PDMS/DVB/CAR) solid-phase microextraction (SPME) extraction head (Supelco, Bellefonte, PA, USA) was inserted. For the GC−MS determination, GC−MS analysis was performed on a GC−MS/MS Quantum TSQ 9000/TRACE 1310 (Thermo Fisher Scientific, Waltham, MA, USA) using a J&W 122-4732 DB-17 ms (30 m × 0.25 mm × 0.25 μm) column. After adsorption on SPME fiber of 50/30 μm DVB/CAR/PDMS (Supelco, Bellefonte, PA, USA) on a magnetic stirrer (Corning, NY, USA) at 50 °C for 30 min, the extraction fiber was loaded into the GC inlet set at 220 °C and desorbed without splitting the helium stream for 2 min. The oven temperature program was as follows: 50 °C, held for 6 min, followed by 3 °C/min up to 120 °C, held for 2 min, 10 °C/min up to 250 °C, held for 2 min. The temperature of the ion source was set at 230 °C and the mass spectra were recorded at 70 eV in the electron impact (EI) ionization mode. The MS range was collected from 29 *m/z* to 540 *m/z* and the scanning rate was 2.88 scan s^−1^. The carrier gas was helium, and the flow rate was 1.0 mL/min. The corresponding peaks of individual compounds were identified based on the NIST/WILEY spectrum library and sample peak areas were identified and quantified relative to the RI and mass spectra of available authentic standards. Each standard was dissolved in methanol to obtain a stock solution. Stocks were diluted in the appropriate aqueous buffer for free volatiles (pH 3.2, citrate–phosphate buffer solution) and bound volatile compounds (pH 5, citrate–phosphate buffer solution), respectively to obtain individual standard curves from between 5 and 9 concentration levels (2.5, 5, 10, 20, 40, 80, 160, 320, 640 ng/g, Appendix A and Appendix A). The standards were extracted under the same conditions as used for the grape samples (Section 3.4).

### 3.6. Assay for TPS (Terpene synthase) Activity

TPS enzyme activity assays were performed using the kit of Tong Wei (TW—reagent. Shanghai, China) according to the manufacturer’s instructions. To characterize the amount of TPS in the sample, a purified antibody specific to plant TPS was first used to coat microtiter plate wells, thereby creating a solid-phase antibody. Subsequently, TPS was then added to the wells and combined with an HRP labeled antibody which enabled the creation of an antibody–antigen–enzyme–antibody complex. After washing completely, TMB substrate solution was added and a blue color was formed when the TMB substrate was catalyzed by the conjugated HRP enzyme. The catalysis reaction was terminated by the addition of a sulfuric acid solution and the color change was measured spectrophotometrically at a wavelength of 450 nm. The concentration of TPS in the samples was then determined by comparing the OD of the samples to the standard curve, as previously reported by [52].

Grape berries were flash frozen in liquid nitrogen and ground to a fine powder. Afterwards, 1 g of the powder was placed in a centrifuge tube containing 5 mL of extraction phosphate buffer (pH 6.8, 0.01 M). The mixture was centrifuged at 3000 rpm for 15 min at 4 °C and the resulting supernatant was collected for the TPS activity assay. Samples were evaluated in a 96-well format using a microplate reader according to the protocol as follows:Addition of standard samples: Standard wells were designated within the plate and 50 µL of the standards were subsequently added to the specified locations;Addition of samples: Blank wells were designated within the plate at specified locations. Transfer of solutions were similar to the process for samples and standards; however, blank wells lacked the addition of samples and the HRP conjugate reagent. For designated sample wells, a total of 40 μL of diluted samples was added to the testing sample wells. Care was taken to not touch the walls of the microplate wells and solutions were gently mixed. With the exception of blank wells, 100 μL of the HRP conjugate reagent was added to each well;Temperature incubation: After sealing the plates with film, samples were incubated at 37 °C for a total of 60 min;Washing: Sealant membranes were carefully removed from plates and washing buffer was added to every well. Thirty seconds later, the solutions were drained and this process was then repeated for a total of 5 times;Coloration: Fifty µL of chromogen solution A and B was added to each well and maintained under dark conditions for a period of 15 min at 37 °C;Reaction termination: 50 μL of stop solution was added to each well, resulting in the development of a yellow color which indicated the termination of the reaction;Assay quantification: Blank wells were taken as zero and absorbance (450 nm) was measured for all samples within 15 min of stopping the reaction.

### 3.7. RNA Isolation and Quantitative Real-Time PCR Analysis

A total of 50–100 mg of frozen berry samples representing the different development stages was ground in a mortar and the resultant powder was used for the RNA extraction with the TRIzol extraction reagent (Invitrogen, Carlsbad, CA, USA). Agarose gel electrophoresis was used to assess the quality of every RNA sample and high-quality RNA samples with an A260/A280 ratio of 1.8:2.1 were used as the template material for the qRT-PCR analyses. Specifically, complementary DNA (cDNA) was transcribed using the Takara Prime Script TM-RT PCR reagent Kit (Takara, Shiga, Japan) according to the manufacturer’s instructions. Since the monoterpene volatiles showed the most significant variations after the elicitor treatment, we quantified the expression of genes in the terpenoid metabolic pathways by qRT-PCR analysis (ABI7300, Applied Biosystems, Milan, Italy). Relative transcript levels were calculated using the 2^−ΔΔCt^ method [53] and *VvActin* was used as the reference for normalization purposes. All reactions, including non-template controls, were performed in triplicate. Sequences of the primers used to amplify the *VvDXS1*, *VvDXS3*, *VvDXR* and *VvHDR* genes were referenced from Martin [43]; *VvPNGer* and *VvPNlinNer1* were referenced from Matarese [47].

### 3.8. Statistical Analysis

All analyses were performed using three biological replicates (biological parameters, aroma compounds, TPS enzyme activity and gene expression level). All of the analyzed data were expressed as means ± standard deviation (SD). The SPSS Version 17.0 (SPSS Inc. Chicago, IL, USA) and Excel software were used to perform the statistical analyses. A one-way ANOVA with a Duncan’s *t*-test was used to evaluate the values (*p* ≤ 0.05).

## 4. Conclusions

In conclusion, grape berries exhibited lower concentrations of free and bound monoterpenes compounds after application with TDZ. Results demonstrate that exogenous application of TDZ decreases the TPS enzyme activity and negatively regulates the expression of several genes involved in the terpene metabolic pathway towards the biosynthesis of monoterpenes. In the MEP pathway, DXS, DXR and downstream TPS were key determinants for monoterpenes biosynthesis. TDZ is an effective elicitor that influences the volatile compounds by altering the MEP and TPS gene expression and resulting enzyme activity within Shine Muscat grapes. These findings provide an opportunity for future research to further explore grape volatile biosynthesis and metabolism at the molecular level by testing different regimes of elicitor application and/or testing different compounds. To the best of our knowledge, this is the first published report to document the effect of TDZ on the quality and quantity of grape volatiles. However, further research pertaining to the physiological and molecular mechanisms related to the application of TDZ on the synthesis of volatiles in grape berries is warranted to gain more insights. This is especially justified because of the impact of volatiles on the aroma and flavor and the links to consumer preference and liking.

## Figures and Tables

**Figure 1 molecules-25-02578-f001:**
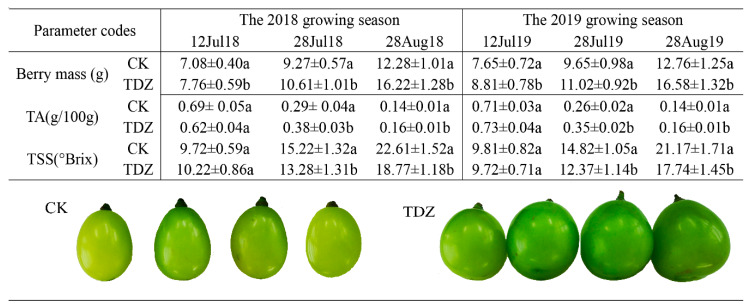
Top: Physico-chemical characteristics of grape berries at three different phenological stages of grape ripening with or without thidiazuron (TDZ) treatment over the 2018 and 2019 growing season. Mean ± Standard error (*n* = 3). Different letters indicate significant differences (Duncan’s multiple range test, *p* ≤ 0.05). CK: control, TDZ: thidiazuron. Bottom: comparison of CK and TDZ appearance (28th Aug., 2018).

**Figure 2 molecules-25-02578-f002:**
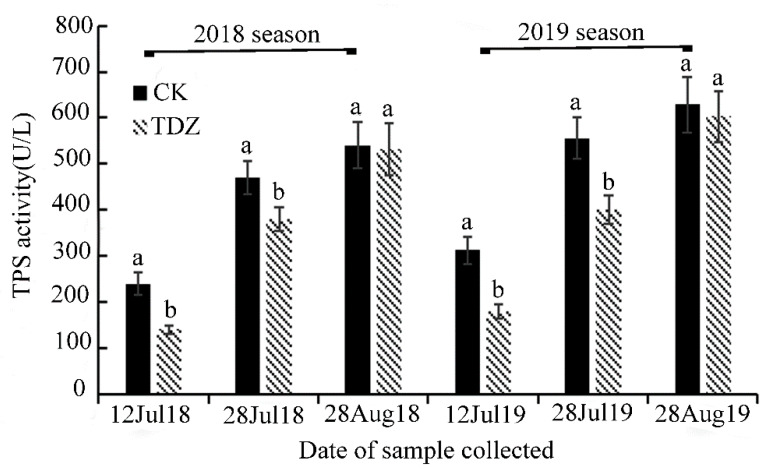
Effect of TDZ on grape terpene synthase (TPS) activity in 2018 and 2019 with the progression of berry development. Mean ± Standard error (*n* = 3). Different letters indicate significant differences (Duncan’s multiple range test, *p* ≤ 0.05).

**Figure 3 molecules-25-02578-f003:**
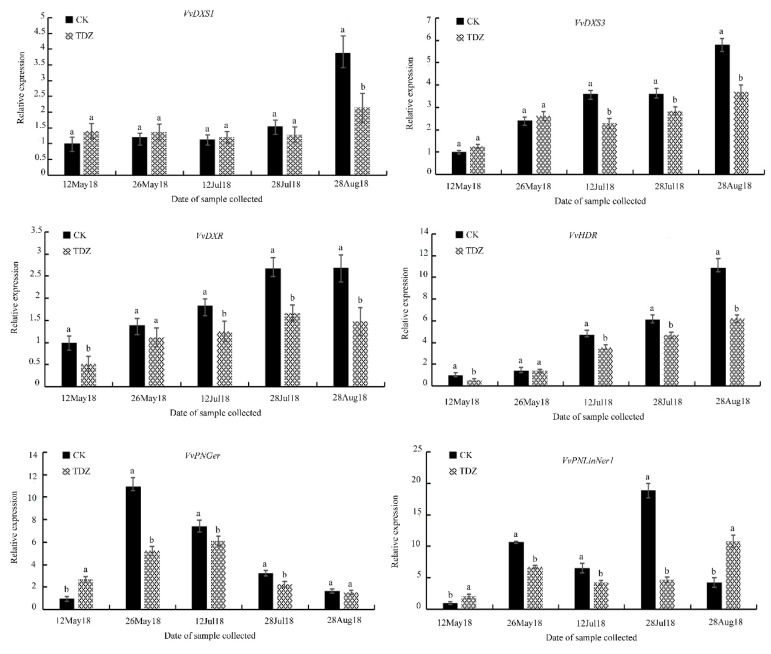
Effect of TDZ on the gene expression of berries in 2018 with the progression of berry development. Mean ± Standard error (*n* = 3). Different letters indicate significant differences (Duncan’s multiple range test, *p* ≤ 0.05).

**Figure 4 molecules-25-02578-f004:**
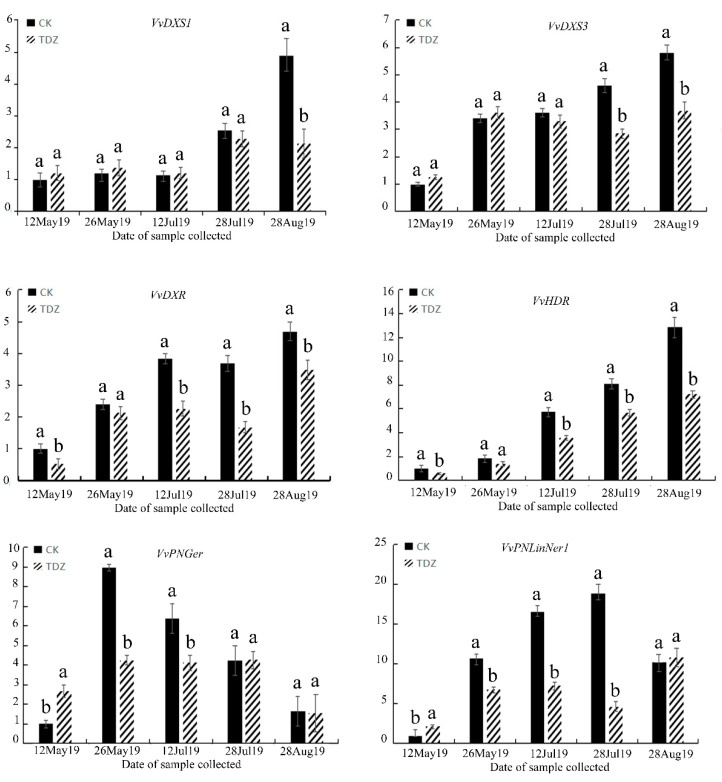
Effect of TDZ on the gene expression of berries in 2019 with the progression of berry development. Mean ± Standard error (*n* = 3). Different letters indicate significant differences (Duncan’s multiple range test, *p* ≤ 0.05).

**Table 1 molecules-25-02578-t001:** Free monoterpene aroma compounds in berries treated with TDZ or control (CK) and sampled at harvest. Mean and standard error (*n* = 3). Different letters indicate significant differences (Duncan’s multiple range test, *p* ≤ 0.05). FW: Fresh weight.

Free Terpene Aroma Compounds (ng/gFW)	Shine Muscat 2018		Shine Muscat 2019	
Monoterpene Compounds	CK	TDZ	CK	TDZ
Linalool	310.45 ± 7.11a	99.15 ± 1.21b	418.42 ± 17.87a	107.74 ± 6.75b
Hotrienol	7.46 ± 0.19a	5.75 ± 0.25b	15.72 ± 1.21a	8.54 ± 0.72b
α-Terpineol	16.83 ± 1.23a	19.21 ± 1.42a	22.56 ± 1.12b	16.15 ± 0.52a
α-Citral	4.71 ± 0.28a	1.94 ± 0.12b	4.81 ± 0.27a	1.66 ± 0.22b
Citronellol	8.42 ± 0.71a	2.41 ± 0.18b	15.3 ± 0.63a	2.55 ± 0.11b
Neral	12.16 ± 0.51a	3.88 ± 0.38b	13.57 ± 0.67a	3.86 ± 0.25b
Geraniol	57.92 ± 4.77a	6.13 ± 0.21b	181.53 ± 9.11a	7.12 ± 0.16b
Geranic acid	3.12 ± 0.22a	2.06 ± 0.12b	4.64 ± 0.33a	2.44 ± 0.16b
Citronellal	1.04 ± 0.08b	2.75 ± 0.24a	1.96 ± 0.13b	5.21 ± 0.37a
β-Myrcene	16.28 ± 0.96a	4.28 ± 0.33b	34.22 ± 2.01a	11.38 ± 0.71b
d-Limonene	71.37 ± 3.85a	21.82 ± 1.52b	56.46 ± 2.02a	15.45 ± 1.03b
Total	509.31	169.28	769.19	180.44

**Table 2 molecules-25-02578-t002:** Bound monoterpenes compounds in berries treated with TDZ or CK and sampled at harvest. Mean and standard error (*n* = 3). Different letters indicate significant differences (Duncan’s multiple range test, *p* ≤ 0.05). FW: Fresh weight.

Bound Terpene Compounds (ng/gFW)	Shine Muscat 2018		Shine Muscat 2019	
Monoterpenes Compound	CK	TDZ	CK	TDZ
Linalool	32.23 ± 1.54a	8.42 ± 0.87b	137.74 ± 9.75a	47.64 ± 5.52b
Hotrienol	47.31 ± 4.24a	11.3 ± 1.89b	72.42 ± 6.16a	24.71 ± 2.72b
α-Terpineol	23.45 ± 3.21a	6.15 ± 0.52b	42.56 ± 5.12a	2.76 ± 0.81b
α-Citral	20.23 ± 1.86a	4.64 ± 0.39b	34.17 ± 2.92a	2.95 ± 0.31b
Citronellol	9.34 ± 0.65a	5.3 ± 0.43b	9.55 ± 0.71a	5.58 ± 0.37b
Neral	45.02 ± 7.01a	13.57 ± 1.67b	107.86 ± 10.55a	16.8 ± 1.96b
Geraniol	130.13 ± 10.14a	5.53 ± 0.11b	218.12 ± 18.06a	16.62 ± 1.12b
Geranic acid	277.4 ± 14.21a	48.39 ± 4.06b	333.32 ± 24.42a	76.57 ± 9.23b
Citronellal	25.52 ± 1.66a	23.42 ± 2.78a	64.35 ± 4.78a	32.81 ± 2.59b
β-Myrcene	10.23 ± 1.08a	3.12 ± 0.22b	12.32 ± 0.24a	1.72 ± 0.12b
d-limonene	74.34 ± 5.62a	12.72 ± 1.04b	122.41 ± 9.97a	17.41 ± 1.32b
Myrtenol	10.43 ± 0.76a	2.3 ± 0.16b	8.89 ± 0.71a	1.02 ± 0.12b
(*E*)-furanoid linalool	342.42 ± 32.06a	82.42 ± 7.02b	245.65 ± 19.32a	57.27 ± 4.76b
(*Z*)-furanoid linalool	341.18 ± 23.22a	112.26 ± 9.25b	303.42 ± 19.87a	84.42 ± 7.68b
Total	1389.23	339.54	1712.78	388.28

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
