# Peer review of "Effect of Thidiazuron on Terpene Volatile Constituents and Terpenoid Biosynthesis Pathway Gene Expression of Shine Muscat (Vitis labrusca × V. vinifera) Grape Berries"

_molecules, 2020, doi:10.3390/molecules25112578_

Round 1
Reviewer 1 Report
The authors have made changes and additions and addressed the points from the first version. There are some minor changes I am suggesting at this point including clarifications and moving some paragraphs in a different section.

Author Response
Thanks you very kindly for the thorough review and please express my thanks to your thoughtful suggestions, I have made changes of the manuscript as your suggest point-by-point, attachment is the revised manuscript. We feel that with these changes the manuscript has improved greatly and may now be considered for publication in your journal.
Reviewer 2 Report
This manuscript was revised as recommended by reviewers. I beliebe that this revised manuscript has been substantially improved and therefore I hope that it can now be published in this journal.
Author Response
Thanks you very kindly for the thorough review and please express my thanks to your kindly acception of our manuscript
Reviewer 3 Report
The authors present a two-year field study, what is the minimum required for this type of analysis. Research results are well correlated with literature. Conclusion. The work is interesting, written in understandable language. The only remark some poems have different spacing, you need to standardize it (see: 3.5. SPME-GC-MS analysis of aroma compounds).
Author Response
Thanks you very kindly for the thorough review and please express my thanks to your kindly acception of our manuscript, the spacing of the paragraph has been standard as your suggestion.